

# Identification and analysis of the expansin gene family in yam

Xing Qiao[1,2,3], Shuai Yuan[1,2,3], Jing Wei[1,2,3], Changzhou Li[1,2,3], Lan Lan[1,2,3], Xuerui He[1,2,3], Aiqin Wang[1,2,3], Jie Zhan[1,2,3], Longfei He[1,2,3] and Dong Xiao[1,2,3]

[1] College of Agriculture, Guangxi University, Nanning, Guangxi Zhuang Autonomous Region, China
[2] National Demonstration Center for Experimental Plant Science Education, Nanning, China
[3] Guangxi University Key Laboratory of Crop Cultivation and Tillage, Nanning, China

## ABSTRACT

Expansins are a group of proteins that loosen plant cell walls and cellulose materials and are involved in regulating plant cell growth and diverse developmental processes. However, a systematic study of the *Dioscorea opposita* expansin (DoEXP) gene family has not yet been conducted. In this study, we used publicly available genomic data from yam to identify members of the DoEXP gene family and analyse its physicochemical properties, phylogeny and expression pattern using bioinformatics software. Thirty EXP genes were identified from the yam genome and can be classified into four subfamilies, DoEXPA, DoEXPB, DoEXLA, and DoEXLB, which are distributed across 14 chromosomes. All EXP proteins contain two conserved structural domains (DPBB_1 and expansin_C) and have highly similar motif compositions and exon–intron structures. Examination of the promoter regions of these genes revealed the presence of *cis*-acting elements associated with growth and development, phytohormone signaling, and abiotic stress. The collinearity analysis revealed that segmental duplication is the primary cause of the expansion of the DoEXP gene family. Gene Ontology (GO) enrichment analysis revealed that DoEXP genes (*DoEXPs*) are significantly associated with root elongation and cell differentiation processes. Utilizing quantitative real-time polymerase chain reaction (qPCR), the expression patterns of nine selected *DoEXPs* were validated. The results of this study are helpful for understanding the molecular functions of expansin proteins in yam tuber expansion and provide a theoretical basis for revealing the molecular regulatory mechanism of yam tuber growth and development.

## BACKGROUND

Expansins are non-enzymatic plant cell wall-loosening proteins involved in cell wall expansion and cell enlargement, and they participate in various developmental processes and responses to abiotic stress. The EXP gene families have been extensively studied in numerous plant species. Although the whole genome sequence of the yam (*Dioscorea opposita*) is available, a comprehensive analysis of its EXP gene family remains lacking.

Corresponding authors
Longfei He, lfhe@gxu.edu.cn
Dong Xiao, xiaodong@gxu.edu.cn

## METHODS

Using bioinformatics tools, expansin genes were identified from the whole genome sequence of yam. A comprehensive analysis of the DoEXP gene family was conducted, including their physicochemical properties, phylogenetic relationships, gene structures, *cis*-acting element analysis, and collinearity analysis. The expression patterns of the *DoEXP*s at different growth and developmental stages were anzlysed using RNA-Seq data obtained in the laboratory, and the results were validated by real-time quantitative polymerase chain reaction (qPCR).

## RESULTS

In this research, 30 *DoEXP*s were identified and classified into four subfamilies, located on fourteen different chromosomes. Gene structure and evolutionary analysis revealed that the high conservation of the expansin gene family in yam is attributed to purifying selection during evolution. The amplification of this family is associated with segmental duplication and tandem duplication. Predictive promoter analysis indicated that DoEXP genes may be responsive to light, hormone signaling, and stress stimulation. RNA-Seq and qPCR analyses obtained and confirmed the expression patterns of *DoEXP*s at different growth stages. This study provides a basis for further functional research and breeding of high-yield yam varieties.

## INTRODUCTION

Yam (*Dioscorea opposita*) is a monocotyledonous plant belonging to the genus *Dioscorea* in the family *Dioscoreaceae*. It is harvested from underground tubers and ranks among the world's top ten food crops. Additionally, it is an important medicinal and food crop in China and is valued for its economic importance (*Zhou et al., 2021*). Its tubers are rich in starch, cellulose, amino acids, sugars, fats, and other nutrients, as well as medicinal components such as allantoin, diosgenin, polysaccharides, and polyphenols (*Obidiegwu, Lyons & Chilaka, 2020*; *Epping & Laibach, 2020*). These components tonify the spleen, nourish the stomach, strengthen the lungs, and benefit the kidneys. The tubers can be used to treat conditions such as prolonged diarrhoea due to deficient spleen function, chronic intestinal inflammation, gastritis, and diabetes mellitus (*Khol et al., 2024*; *Zhang et al., 2024*). The growth and development of tubers affect the yield and quality of yam, and the function of expansins is closely related to the expansion of yam tubers (*Marowa, Ding & Kong, 2016*).

Expansins (EXPs) are cell wall-associated proteins that participate in cell wall loosening and cell enlargement in a pH-dependent manner (*Cosgrove, 2000*). Expansins are ubiquitously found in various plants (*Wu, Meeley & Cosgrove, 2001*; *Lin et al., 2005*; *Shi et al., 2014*; *Gao et al., 2020*; *Li et al., 2024a*). In plants, expansins are involved in cell wall relaxation. The growing cell wall is structured as a multi-layered arrangement of nearly parallel cellulose microfibrils tethered by hemicellulose. Expansins mediate the disruption of these microfibrils, leading to cell wall relaxation and facilitating cell elongation (*Marga et al., 2005*). The expansin superfamily can be divided into four

families: α-expansins (EXPA), β-expansins (EXPB), expansin-like A (EXLA), and expansin-like B (EXLB). Typically, expansins consist of 250–275 amino acids, including two conserved domains, DPBB_1 and expansin_C, and a signal peptide (SP) region located at the N-terminus with a length of 20–30 amino acids. Domain 1 is a conserved region featuring a double-psi β-barrel (DPBB) fold located in the catalytic domain in the middle of the entire protein sequence. It contains 120–135 amino acids, is rich in cysteine residues, and shares high homology with glycoside hydrolase family 45 (GH45). The expansin_C domain is a stable domain located at the carboxyl terminus of expansins and generally contains 90–120 amino acids. This domain has a certain degree of homology with Group 2 grass pollen allergen proteins (G2A) in *Poaceae* (*Sampedro & Cosgrove, 2005*).

Expansins are involved in plant growth and abiotic stress responses (*Choi et al., 2003*; *Jin et al., 2020*). Research indicates that the aluminium-induced expansin gene *OsEXPA10* contributes to rice root cell elongation (*Che et al., 2016*). Enhanced expression of *AcEXPA23* drives lateral root development in kiwifruit (*Actinidia chinensis Planch.*) (*Wu et al., 2022*). Transgenic overexpression of *TaEXPA7-B* in rice enhances salt tolerance and stimulates lateral root primordia and cortical cell development (*Wang et al., 2024a*). *HvEXPB7* elevates drought tolerance in barley (*Hordeum vulgare L.*) by boosting root hair growth under drought stress (*He et al., 2015*). Overexpression of *TaEXPA2* improves seed yield production and drought tolerance in tobacco (*Chen et al., 2016*). *AtEXPA18* enhances drought tolerance in transgenic tobacco plants (*Abbasi, Malekpour & Sobhanverdi, 2021*). *CqEXPA*50 promotes salt tolerance *via* interactions with the auxin pathway in quinoa (*Chenopodium quinoa*) (*Sun et al., 2022*). Overexpression of *GsEXLB14* enhances salt and drought tolerance in soybean hairy roots (*Wang et al., 2024b*). *AcEXPA1* overexpression elevates aluminium resistance in *Axonopus compressus via* root growth regulation (*Li et al., 2024b*).

Collectively, these findings advance our comprehension of expansin functions, highlighting their important roles in both developmental processes and stress responses. However, the understanding of expansin characteristics—including family composition, expression dynamics, and functional attributes—in yam remains limited. This study identified and analysed EXP family members in yam using genome database, laying a foundational framework for future investigations into their roles in growth regulation and stress adaptation mechanisms in yam.

## MATERIALS AND METHODS

### Identification of *DoEXP*s, analysis of physicochemical properties, and construction of phylogenetic trees

The genome file (Dalata_550_v2.1) was retrieved from the Phytozome database (https://phytozome-next.jgi.doe.gov/) to access the yam genome and protein sequences, enabling the identification of *DoEXP*s. A two-step approach was employed to characterize the EXP gene family in yam. First, the amino acid sequences of 35 EXPs were retrieved from the Arabidopsis database (https://www.arabidopsis.org/) and utilized as query sequences. A genome-wide screening of yam was performed using TBtools with an E-value threshold of

$1 \times 10^{-5}$. Subsequently, conserved domains of Arabidopsis EXP proteins were identified *via* PFAM (http://pfam.xfam.org/), revealing that all yam EXPs (DoEXPs) contain the DPBB_1 (PF03330) and expansin_C (PF01357) domains. Second, hidden Markov models (HMMs) based on PFAM domains PF03330 and PF01357 were constructed to conduct HMMsearch analyses on the yam protein dataset (E-value ≤ $1 \times 10^{-5}$). Candidate genes from both methodologies were merged, with overlapping entries filtered. The final yam expansin protein sequences were subjected to computational analysis: molecular weight (MW), isoelectric point (pI), and hydrophilicity were calculated using ExPaSy (https://web.expasy.org/compute_pi/), while subcellular localization predictions were generated *via* Plant-mPLoc (http://www.csbio.sjtu.edu.cn/bioinf/plant-multi/).

Amino acid sequences were aligned using ClustalW (*Thompson, Higgins & Gibson, 1994*). The phylogenetic tree was generated in ClustalW by the neighbour-joining method and a thousand replicates and displayed using iTOL (https://itol.embl.de/itol.cgi) (*Letunic & Bork, 2024*).

## Analysis of conserved protein motifs, stable domains, and gene structures of *DoEXP*s

The MEME Suite (https://meme-suite.org/meme/tools/meme) was employed to identify conserved motifs existing in the 30 yam *EXP*s using default parameters, with the number of motifs specified to 10. Following motif identification, TBtools was utilized to visualize the retrieved conserved protein motifs. Conserved domains and their genomic positions in *DoEXP*s were retrieved from the InterPro database (https://www.ebi.ac.uk/interpro/). Domains annotated with PFAM accession numbers were used as screening criteria, and the corresponding domain data were manually curated. Subsequently, TBtools was employed to visualize the curated domain distribution. Exon-intron structures of the genes were diagrammed using genomic annotation data from the yam genome database.

## Chromosomal localization of the genes and analysis of *cis*-acting elements in the promoters of *DoEXP*s

The yam genome annotation data were employed to anzlyse the chromosomal localization of the renamed yam EXP genes with TBtools. The bin size was set to 100 kb, and the gene density data files for each chromosome were exported and visualized in the chromosome frames in the figure.

The genomic sequence 2,000 bp upstream of the target gene was extracted using TBtools. The promoter sequences and the corresponding genomic data of yam were anzlysed for the presence of response elements using the Plantcare online tool (http://bioinformatics.psb.ugent.be/webtools/plantcare/html/). The analysis was performed to extract *cis*-acting elements in the promoters related to growth and development, phytohormones, and the stress response using the output of the website, and the data were subsequently visualized using TBtools. The number of major *cis*-acting elements contained in each EXP gene was quantified, and a heatmap was constructed.

## Covariance analysis of *EXP*s

Intraspecific collinearity analysis: The McscanX module within TBtools was utilized to investigate tandem and segmental duplications among *DoEXP*s. The bin size was set to 100 kb to obtain the gene densities on chromosomes. For each gene occurrence, a value of 1 was assigned, and occurrence counts were aggregated. Heatmaps and line graphs were generated to depict the gene density on each chromosome. Subsequently, collinear relationships among *DoEXP*s were visualized *via* the Advanced Circos module. Tandem duplications were defined as nucleotide sequences with ≥75% alignment identity/similarity and intergenic distances <100 kb on the same chromosome. Segmental duplications were identified as genes located in duplicated regions with nucleotide alignment rates ≥75%. These criteria followed the methodology of *Cannon et al. (2004)*. Gene pairs derived from tandem duplication and segmental duplication were subjected to selection pressure analysis using TBtools' Simple Ka/Ks Calculator (NG). The nonsynonymous-to-synonymous substitution rate ratio (Ka/Ks) was calculated. Genes were classified as undergoing positive selection (Ka/Ks > 1), neutral evolution (Ka/Ks = 1), or purifying selection (Ka/Ks < 1). The Ks value was used to estimate the approximate date of each duplication event that occurred in yam with the following formula: T = Ks/2λ × $10^{-6}$ Mya (million years ago) (λ = 6.5 × $1^{-9}$), as described in *Chang et al. (2023)*.

Interspecific collinearity analysis: Collinearity analysis was performed among three species, including the monocotyledonous plant rice, the dicotyledonous plant Arabidopsis, and the monocotyledonous plant yam. The genomic and gene annotation data for rice were retrieved from the Ensembl Plants database (https://plants.ensembl.org/index.html). The genomic and annotation datasets for yam were sourced as previously described in this article. Interspecific collinearity analysis was conducted using the McScanX module within TBtools with default parameters. DoEXP genes exhibiting collinear relationships with orthologs from the other two species were identified and highlighted for visualization. Moreover, collinear gene pairs between yam and rice as well as between yam and Arabidopsis were identified, the evolutionary selection pressure values (Ka/Ks) were calculated, and scatter plots were generated using ggplot2.

## Heatmap and analysis of the expression of *DoEXP*s

The transcriptome data of the existing varieties in the laboratory were used as the source data to analyse the expression levels of *DoEXP*s. These data were obtained by measuring the gene expression levels of the yam variety "NH1" (abbreviated as NH) in four growth stages, with three technical replicates per stage to ensure data reliability. Gene expression levels were quantified as FPKM values. Subsequently, the expression levels of the target genes in this study, which were measured three times in each of the four periods, were collated and summarized. Given the absence of detectable expression for *DoEXPA14* and *DoEXPA15* in transcriptomic data, these genes were excluded from subsequent visualization. Finally, the HeatMap module of TBtools was applied to visualize expression dynamics. Using its built-in row normalization and Euclidean clustering methods, the

trends for the changes in the expression levels of each gene in different periods were displayed, and the genes with similar expression patterns were clustered.

## GO functional enrichment analysis of *DoEXP*s

The yam genome was annotated at the genome-wide level using the EggNOG-mapper (http://eggnog-mapper.embl.de/). Gene Ontology (GO) enrichment analysis was performed using the R package clusterProfiler. Pathways with $P < 0.05$ were identified as significantly enriched, and enrichment bubble plots were generated using ggplot2.

## Plant materials

The experimental material used in this study was the "NH1" variety, cultivated at the Guangxi University Agricultural Demonstration Base. In late April 2024, healthy seedlings with uniform growth that were 30 days old and 15–20 cm in height were selected for transplanting, with a plant spacing of 30 cm and row spacing of 180 cm. Four months after planting, the tubers entered the formation stage (NH-F), at which point the first sampling was conducted. Subsequent samplings were performed monthly, covering the following stages: early tuber swelling (NH-E), middle tuber swelling (NH-M), and late tuber swelling (NH-L). To ensure sample representativeness, a stratified random sampling method was employed in this study. At each developmental stage, 2–3 tuber samples were randomly collected from three distinct plots within the experimental field. The same anatomical segment of each tuber was excised, peeled, and sliced into thin sections using a sterile blade. Sections from three randomly selected tubers were pooled to form one experimental replicate, with three or more biological replicates per stage. Upon collection, samples were immediately flash-frozen in liquid nitrogen; once fully frozen, they were stored at −80 °C for subsequent analyses.

## RNA extraction, cDNA synthesis, qPCR, and expression analysis

The Vazyme FastPure Universal Plant Total RNA Isolation Kit (Vazyme, Nanjing, China) was used for RNA extraction, with all procedures strictly following the manufacturer's instructions. RNA integrity was confirmed by 1.2% agarose gel electrophoresis, which was visualized under UV light and displayed clear 28S and 18S rRNA bands, indicating intact RNA. RNA concentration and purity were quantified using a NanoDrop ONE spectrophotometer (Thermo Fisher Scientific, Waltham, MA, USA), and samples with concentration ≥150 ng/µL, 260/280 nm absorbance ratios between 1.8–2.0, and 260/230 nm absorbance ratios ≥2.0 were considered acceptable. First-strand cDNA synthesis was performed using 1.0 µg of qualified total RNA and the HiScript III All-in-one RT SuperMix Perfect for qPCR Kit (Vazyme, Nanjing, China) following the manufacturer's protocol. The diluted cDNA product served as the template for qPCR. *DoActin* (GenBank accession no. KU669295) served as the reference gene. The relative expression level of genes was calculated by the $2^{-\Delta\Delta Ct}$ method (*Livak & Schmittgen, 2001*). qPCR analysis was performed on a BIO-RAD CFX96™ Real-Time System (Bio-Rad, Hercules, CA, USA), following the manufacturer's instructions. The reaction consisted of 5 µL AceQ qPCR SYBR Green Master Mix (Vazyme, Nanjing, China), 0.2 µmol/L upstream and downstream primers, 1.0

μL cDNA, and up to 10 μL with ddH$_2$O. Three independent repeats were done to give the typical results shown here.

## RESULTS AND ANALYSIS

### Identification and evolutionary analysis of the *DoEXP*s

In this study, 30 *DoEXP*s were identified from the yam genome. These genes were classified into four subfamilies and named sequentially according to their chromosomal locations. Similar to those in other plants, the EXPA subfamily constituted the largest branch with 21 members, followed by four EXLA genes, three EXLB genes, and two EXPB genes. Their genomic coordinates, strand orientations, and physicochemical properties are summarized in Table 1. The encoded proteins exhibited molecular weights (MWs) ranging from 22,016.99 Da (*DoEXPA12*) to 31,599.85 Da (*DoEXPA2*), with amino acid lengths varying from 206 residues to 289 residues. The predicted isoelectric points (pIs) varied from 4.43 (*DoEXLB1*) to 9.66 (*DoEXPA4*), with five expansins having theoretical pIs less than 7, classifying them as acidic proteins, whereas the remaining 25 expansins were alkaline proteins. The hydrophilicity values were calculated based on the amino acids of the proteins. Nineteen expansin proteins displayed negative hydrophilicity values, indicating hydrophilic properties, with *DoEXPA3* being the most hydrophilic. Eleven expansins had positive hydrophilicity values, indicating that they were hydrophobic proteins, and *DoEXPA13* was the most hydrophobic. All proteins were predicted to be localized to the cell wall, consistent with their canonical role in cell wall loosening. Detailed physicochemical properties are summarized in Table 1. A phylogenetic tree was constructed using 152 expansin proteins (34 from Arabidopsis, 30 from yam, 37 from rice, 18 from wheat, and 33 from potato) *via* maximum likelihood analysis (Fig. 1). The expansin genes of different species were clearly divided into four subfamilies, EXPA, EXPB, EXLA, and EXLB, and the members of each of the four subfamilies clustered together.

### Analysis of conserved protein motifs, stable structural domains and DoEXP genes' structures

The MEME suite was used to analyse the amino acid sequences of *DoEXP*s for conserved protein motifs, and the number of motifs was set to 10. The identified conserved motifs are shown in Table S1. Members of the same subfamily exhibited similarities in the type, number, and order of conserved motifs, while divergence was observed between different subfamilies (Fig. 2). The EXPA subfamily, the largest group, exhibited a stable motif composition—consisting of motifs 5, 1, 10, 3, 2, 7, 4, and 6—with the exception of five genes: *DoEXPA11*, *DoEXPA12*, *DoEXPA14*, *DoEXPA19*, and *DoEXPA20*. The EXPB subfamily uniformly contained motifs 5, 1, 8, 7, 4, and 6. Notably, the EXLA and EXLB subfamilies shared the unique motif 9, implying a closer evolutionary relationship between these two subfamilies compared to EXPA and EXPB. Additionally, all EXP genes possessed two conserved motifs, motif 1 and motif 6, indicating that these two motifs remained highly conserved throughout expansin evolution. The two conserved domains were localized to the central and C-terminal regions of the protein sequences. The exon-intron

**Table 1 Details of the identified expansin genes in yam.**

| Gene ID | Gene name | Start | End | Strand | Length (aa) | PI | MW (Da) | Hydrophilicity | Subcellular location |
|---------|-----------|-------|-----|--------|-------------|-----|---------|----------------|----------------------|
| Dioal.01G060200.1.p | DoEXPA1 | 24298694 | 24300058 | – | 258 | 8.54 | 28,087.72 | −0.06 | Cell wall |
| Dioal.05G195500.1.p | DoEXPA2 | 23425696 | 23427143 | – | 249 | 8.12 | 26,906.08 | −0.077 | Cell wall |
| Dioal.06G001700.1.p | DoEXPA3 | 128847 | 129817 | – | 259 | 9.37 | 28,480.21 | −0.244 | Cell wall |
| Dioal.07G093100.1.p | DoEXPA4 | 21043371 | 21045031 | – | 250 | 7.52 | 26,492.72 | −0.026 | Cell wall |
| Dioal.07G106800.1.p | DoEXPA5 | 22165321 | 22167293 | – | 256 | 6.86 | 27,441.79 | 0.034 | Cell wall |
| Dioal.07G127900.1.p | DoEXPA6 | 23506033 | 23508052 | – | 259 | 9.45 | 27,882.7 | −0.046 | Cell wall |
| Dioal.09G059800.1.p | DoEXPA7 | 18651741 | 18653159 | – | 252 | 8.12 | 26,823.91 | −0.087 | Cell wall |
| Dioal.09G077500.1.p | DoEXPA8 | 19835585 | 19841937 | – | 275 | 8.06 | 30,361.68 | 0.056 | Cell wall |
| Dioal.11G043900.1.p | DoEXPA9 | 12662460 | 12663794 | + | 289 | 8.78 | 31,599.85 | −0.017 | Cell wall |
| Dioal.12G089000.1.p | DoEXPA10 | 19701720 | 19703208 | – | 259 | 9.32 | 27,804.85 | 0.055 | Cell wall |
| Dioal.14G031400.1.p | DoEXPA11 | 2723362 | 2726278 | – | 267 | 9.66 | 28,910.06 | 0.024 | Cell wall |
| Dioal.14G121100.1.p | DoEXPA12 | 19925968 | 19926962 | + | 272 | 9.17 | 29,188.13 | −0.107 | Cell wall |
| Dioal.14G128000.1.p | DoEXPA13 | 20458705 | 20460966 | – | 261 | 8.59 | 28,028.25 | 0.129 | Cell wall |
| Dioal.15G009800.1.p | DoEXPA14 | 662455 | 663986 | + | 230 | 6.8 | 24,822.82 | −0.243 | Cell wall |
| Dioal.15G009900.1.p | DoEXPA15 | 665820 | 667145 | + | 254 | 9.44 | 27,870.21 | −0.104 | Cell wall |
| Dioal.16G093500.1.p | DoEXPA16 | 23322610 | 23324229 | + | 271 | 8.05 | 29,706.79 | 0.093 | Cell wall |
| Dioal.18G071400.1.p | DoEXPA17 | 24445322 | 24446987 | – | 251 | 9.11 | 26,627 | 0.024 | Cell wall |
| Dioal.18G109200.1.p | DoEXPA18 | 26752627 | 26754932 | – | 260 | 9.34 | 27,853.68 | −0.007 | Cell wall |
| Dioal.19G023600.1.p | DoEXPA19 | 1540636 | 1541727 | + | 206 | 8.61 | 22,016.99 | 0.011 | Cell wall |
| Dioal.19G082500.1.p | DoEXPA20 | 16501589 | 16503014 | + | 248 | 9.3 | 26,617.22 | 0.002 | Cell wall |
| Dioal.19G119800.1.p | DoEXPA21 | 20283831 | 20285183 | – | 252 | 8.4 | 27,128.46 | −0.098 | Cell wall |
| Dioal.04G180000.1.p | DoEXPB1 | 23997238 | 23998430 | + | 277 | 6.41 | 29,230.91 | −0.049 | Cell wall |
| Dioal.19G078100.1.p | DoEXPB2 | 15631278 | 15634965 | – | 286 | 8.05 | 31,344.75 | −0.1 | Cell wall |
| Dioal.02G029200.1.p | DoEXLA1 | 4063537 | 4064899 | – | 269 | 7.99 | 29,150.32 | −0.025 | Cell wall |
| Dioal.02G029300.1.p | DoEXLA2 | 4067080 | 4068322 | – | 264 | 9.07 | 28,691.12 | −0.046 | Cell wall |
| Dioal.02G029800.1.p | DoEXLA3 | 4128602 | 4130133 | + | 264 | 9 | 28,604.88 | 0.046 | Cell wall |
| Dioal.14G049100.1.p | DoEXLA4 | 7116372 | 7118655 | – | 263 | 8.47 | 28,761.91 | −0.011 | Cell wall |
| Dioal.04G166400.1.p | DoEXLB1 | 23181553 | 23182937 | – | 257 | 4.82 | 27,356.81 | 0.004 | Cell wall |
| Dioal.04G171000.1.p | DoEXLB2 | 23445733 | 23447854 | + | 248 | 7.43 | 27,458.34 | −0.17 | Cell wall |
| Dioal.19G021600.1.p | DoEXLB3 | 1411885 | 1414271 | + | 254 | 4.43 | 27,227.49 | −0.07 | Cell wall |

structure, serving as the structural framework of genes, plays a critical role in modulating gene expression. Results revealed that most subfamily members exhibited comparable exon-intron composition (Fig. 2). Genes in the EXPA subfamily contained 2–3 exons, with three genes (10%) having two exons and 60% possessing three exons. The EXPA subfamily harbored 1–2 introns, whereas the remaining subfamilies possessed 3–4 introns, with exon counts ranging from 4–5. Notably, all EXPB subfamily members contained four exons, and among the seven genes in EXLA and EXLB, six (20%) exhibited five-exon architectures. All percentages are based on the total 30 DoEXPs. Collectively, these findings suggest that genes within each subfamily exhibit a high degree of structural conservation.

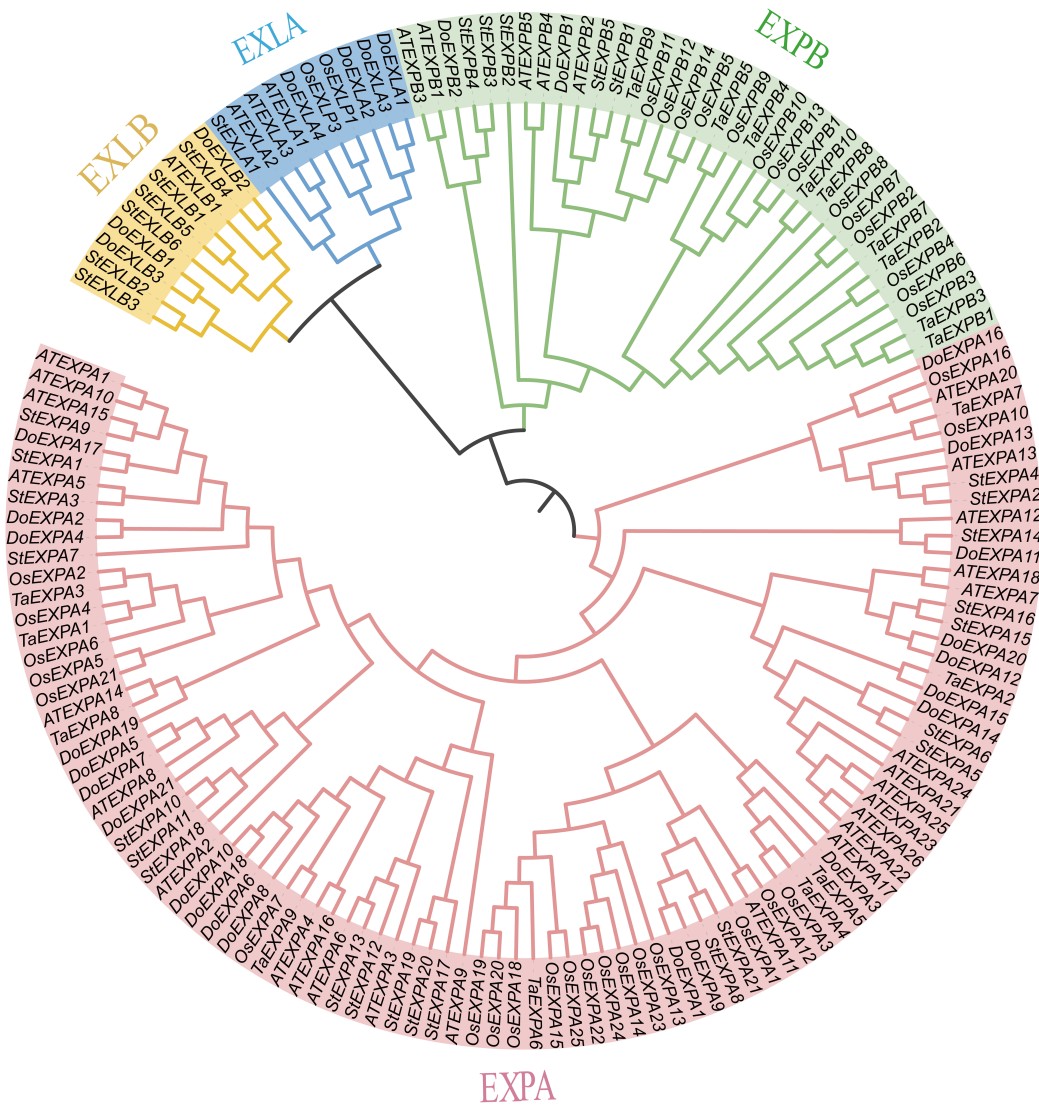

**Figure 1 Phylogenetic evolutionary tree analysis.** A phylogenetic tree constructed using EXPs from several species. The four subfamilies of the EXP family are designated as EXPA, EXPB, EXLA, and EXLB.

## Chromosomal localization of *DoEXP*s and analysis of *cis*-acting elements

The chromosomal distribution of *DoEXP*s was analysed and visualized using TBtools software (Fig. 3). The results indicated that *EXP*s were distributed across all 14 chromosomes, with chromosome 19 harbouring the highest number (five genes: three EXPA, one EXPB, and one EXLB). Chromosome 14 contained four EXP genes, while the remaining chromosomes carried one to three EXP genes. Gene density heatmaps indicated a preferential distribution of EXP genes at chromosomal termini, with the lowest density being observed in central regions. *DoEXLA1-DoEXLA2* on chromosome 2 and

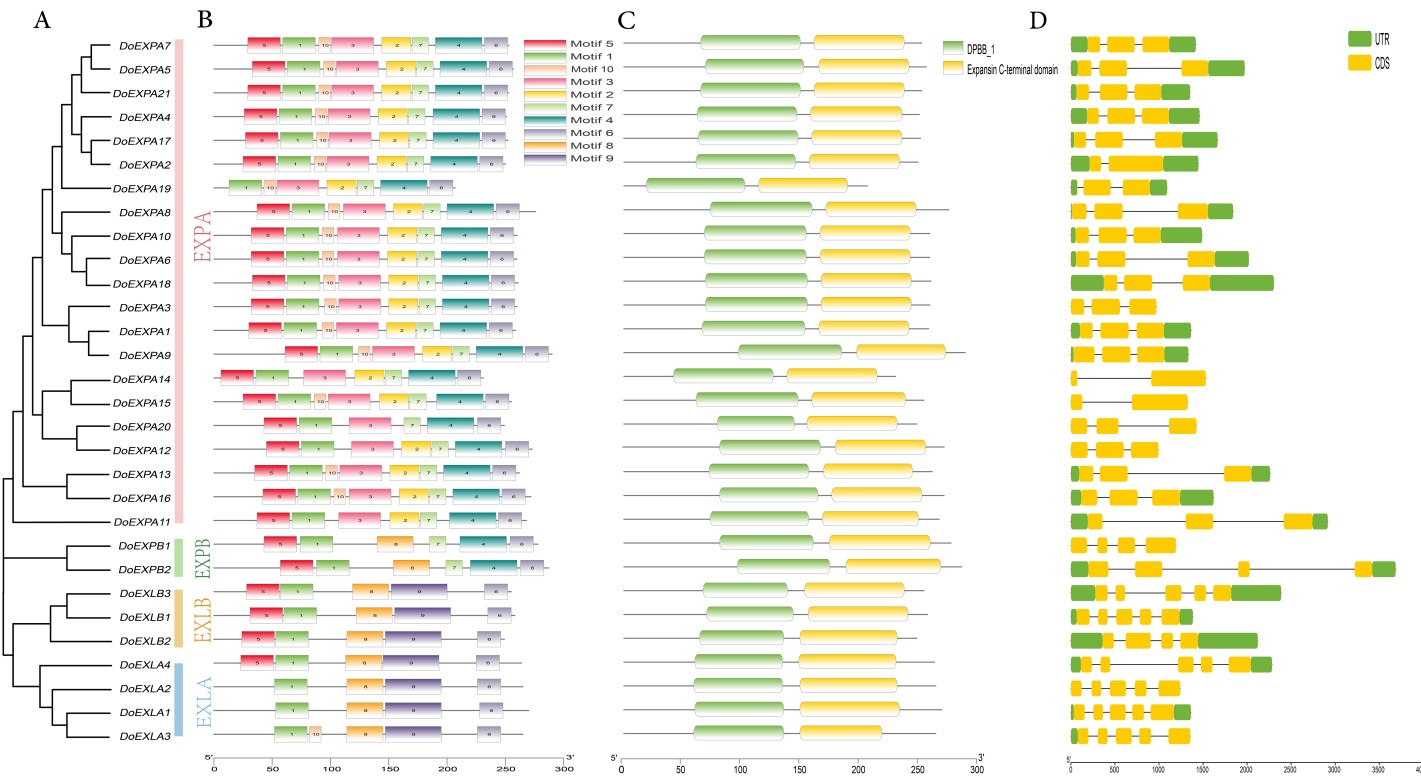

**Figure 2 Phylogenetic relationships, motif composition, stable structural domains, and gene structures of *DoEXPs*.** (A) The phylogenetic relationships of *DoEXPs* are categorized into four groups: α-expansin (EXPA), β-expansin (EXPB), expansin-like A (EXLA), and expansin-like B (EXLB), represented by pink, green, yellow, and blue, respectively; (B) MEME website analysis of the motif composition of *DoEXPs*, with different motifs indicated by different colors; (C) Interpro database to unlock the conserved structural domains and positional information of *DoEXPs*; (D) Structural analysis of *DoEXPs*, with intron and exon structures shown.

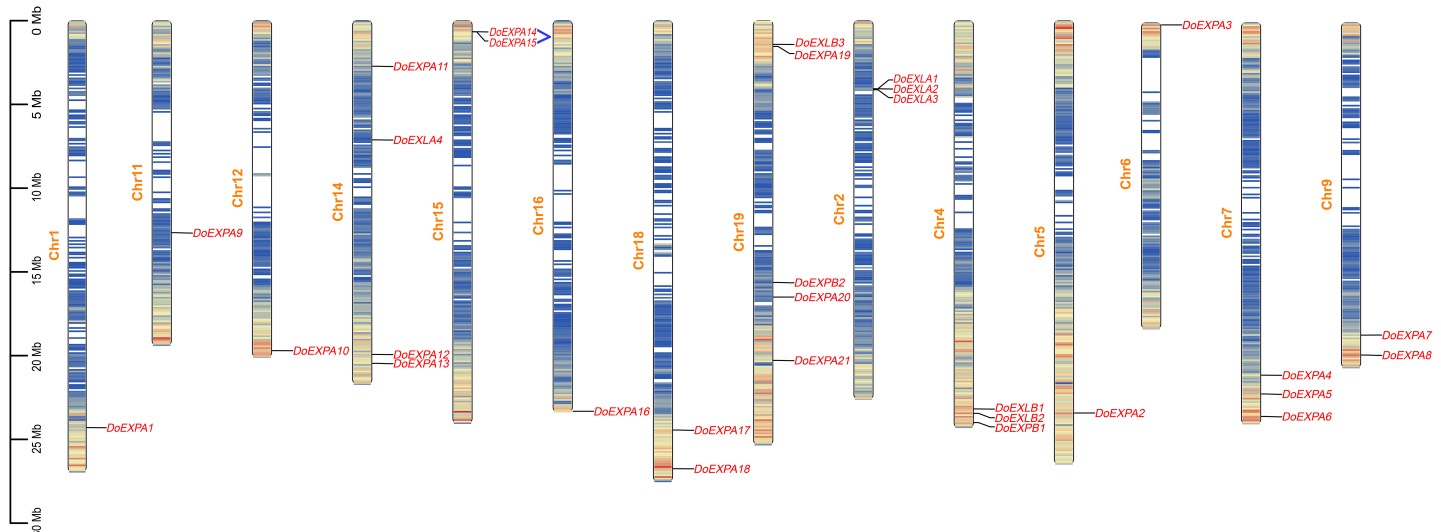

**Figure 3 Chromosome mapping diagram of *DoEXPs*.** Gene density is displayed on chromosomes in the form of a heat map, with high density in red and low density in blue. Two pairs of tandem duplication gene pairs are represented by blue lines.

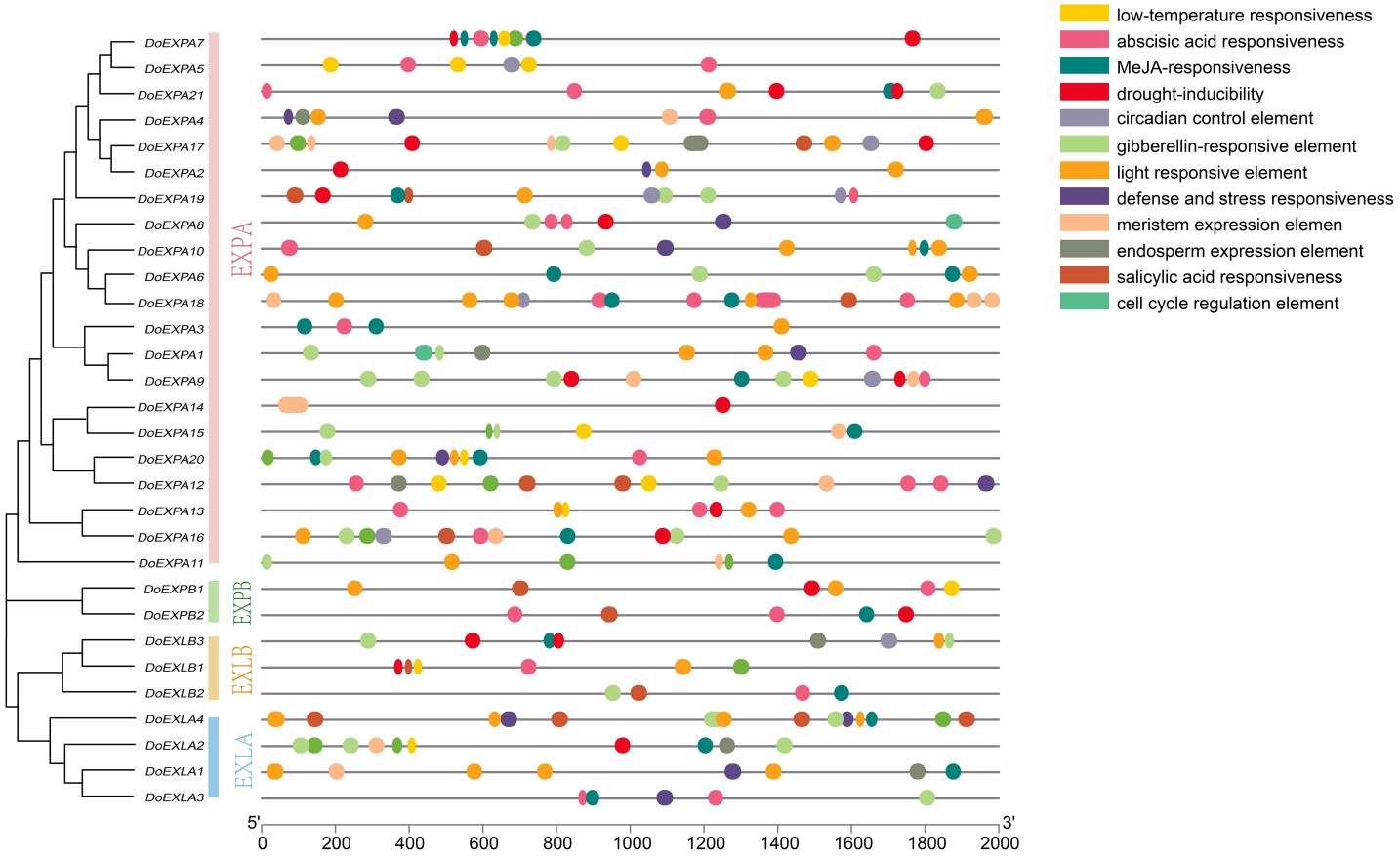

**Figure 4 Cis-acting element analysis of *DoEXP*s.** Cis-acting elements in the 2,000 bp upstream promoters of DoEXP genes were predicted using the Plantcare online tool. Different colored circles represent distinct cis-acting elements.

*DoEXPA14-DoEXPA15* on chromosome 15 were identified as tandemly duplicated gene pairs.

The *cis*-acting elements in the 2,000 bp upstream promoter regions of 30 *DoEXP*s were analysed to elucidate the potential functions and regulatory mechanisms of the DoEXP family genes. A total of 13 major *cis*-acting elements were detected, including light-, hormone-, and abiotic stress-responsive elements. As shown in Fig. 4, 20 genes contained light-responsive elements. The number of key *cis*-acting elements was counted and the results are shown in Fig. 5, where the number of light-responsive elements was the greatest. Additionally, numerous plant hormone-responsive elements were detected, including 38 methyl jasmonate (MeJA)-response elements. MeJA responsiveness is primarily associated with plant defense mechanisms and abiotic stress tolerance. Notably, *DoEXPA3*, *DoEXPA6*, and *DoEXPA18* each harbored four MeJA-responsive elements. A total of 37 abscisic acid (ABA) response elements were identified, with *DoEXPA18* containing four ABA-responsive elements and *DoEXPA12/DoEXPA13* each possessing three. Meanwhile, 31 gibberellin (GA), 16 salicylic acid (SA) and 12 auxin (IAA) response elements were also identified.

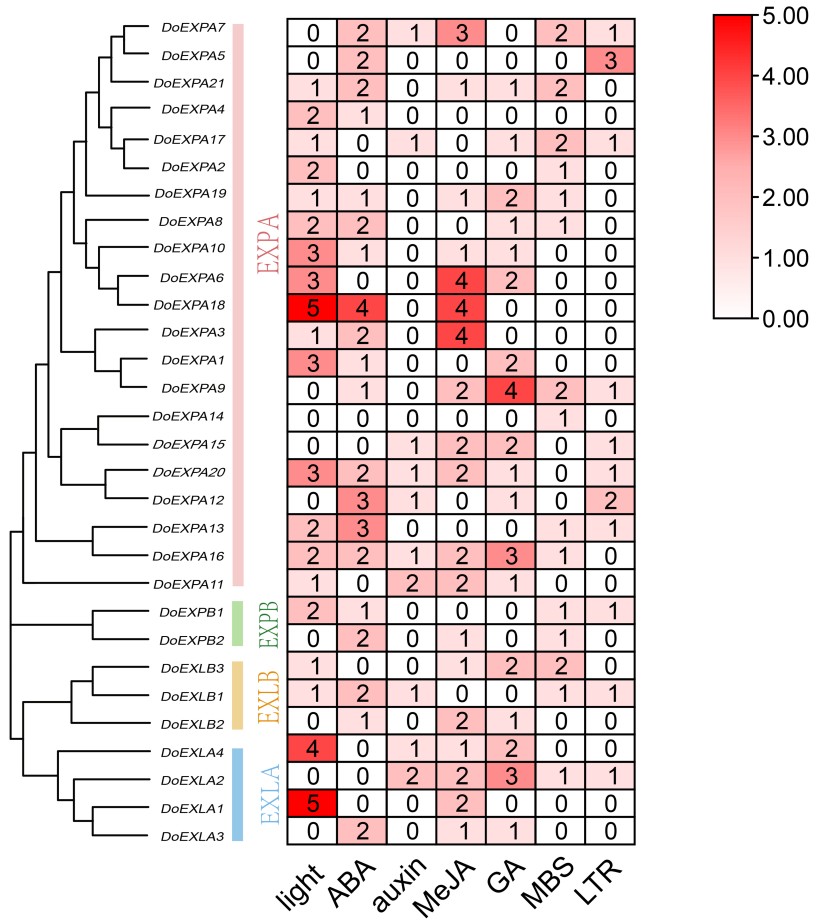

**Figure 5** **Heat map of *cis*-acting element counts in *DoEXP*s promoter regions.** Seven key cis-acting elements of *DoEXP*s are represented by differently coloured boxes with numbers.

In terms of abiotic stress, 20 drought-inducible (MBS) and 14 low-temperature-responsive (LTR) MYB binding sites were identified. Seven genes—*DoEXPA7*, *DoEXPA9*, *DoEXPA13*, *DoEXPA17*, *DoEXPB1*, *DoEXLA2*, and *DoEXLB1*—harbored both these elements, suggesting their potential roles in abiotic stress adaptation. Additionally, eight circadian rhythm control elements, 16 meristem expression elements, seven endosperm expression elements, and two cell cycle regulation elements were also detected. These functionally clustered elements collectively influence plant morphology.

## Intraspecific and interspecific collinearity of *DoEXP*s

Segmental duplication and tandem duplication are the main mechanisms driving the expansion of plant gene families. Collinearity analysis was conducted for *DoEXP*s and between yam and rice, as well as yam and Arabidopsis, to investigate the evolutionary relationships within the EXP gene family. A total of two tandemly duplicated gene pairs and 16 segmentally duplicated gene pairs were identified for *DoEXP*s, with duplication
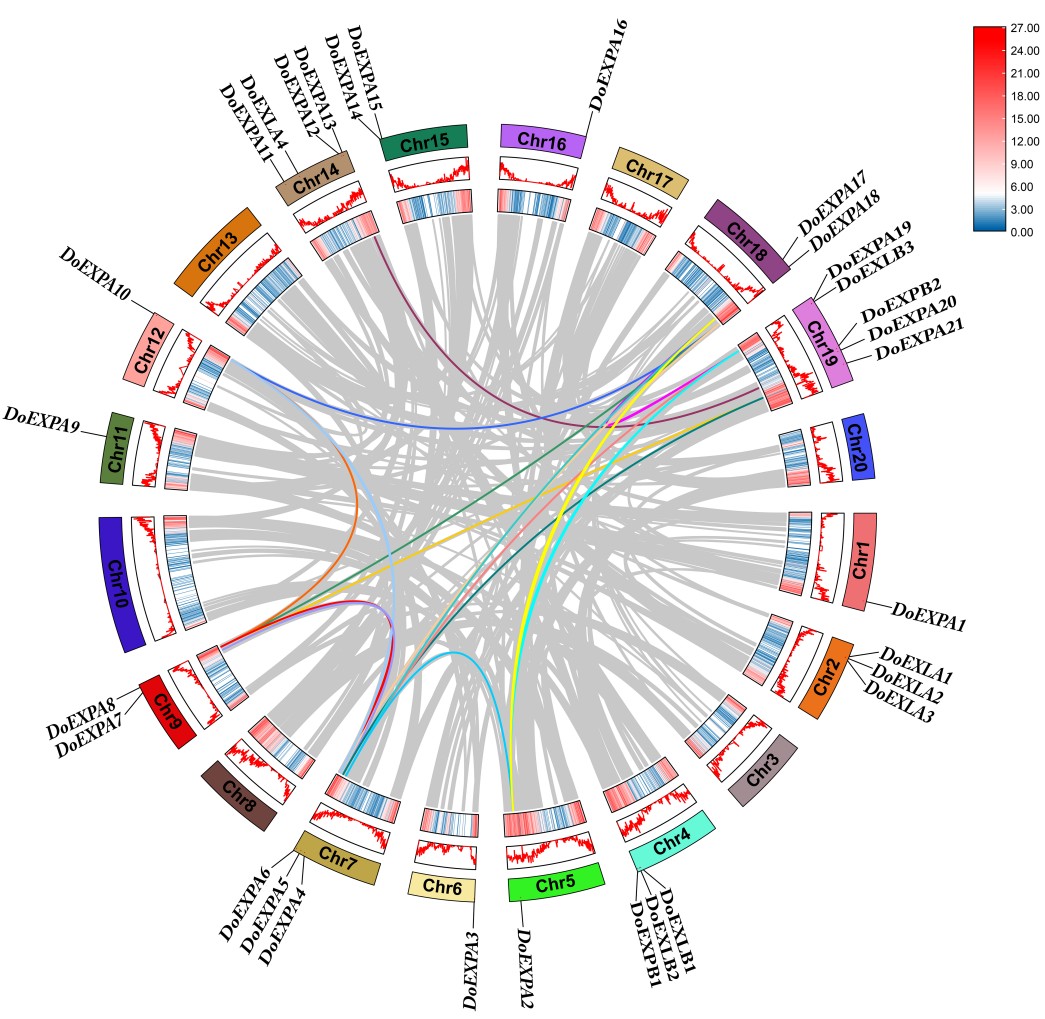

**Figure 6 Intraspecific collinearity analysis of *DoEXPs*.** From the outside to the inside, there are gene names, chromosome names, a line graph of gene density, a heat map of gene density, and the collinearity situation among genes in sequence. The gray lines refer to the gene collinearity information within the whole genome of yam.

data summarized in Table S2. These findings suggest that large-scale chromosomal segmental duplications primarily drove the evolution of DoEXPA genes.

Intraspecific collinear relationships are shown in Fig. 6, where the grey lines represent all collinear gene pairs in the yam genome. For the four subfamilies of *DoEXPs*, collinear gene pairs were observed only within the EXPA subfamily, and no collinear relationships were detected between genes in the other three subfamilies. Chromosome 7 exhibited the highest number of collinearities (eight), followed by chromosomes 18 and 19 (six each). Chromosome 9 had five collinear gene pairs, while chromosomes 5, 12, and 14 contained three, three, and two collinear pairs, respectively.

The Ka/Ks ratios were calculated for all segmental duplication pairs except *DoEXPA1–DoEXPA4*, for which the ratio was uncomputable (Table 2). The remaining 17 duplicated gene pairs all had Ka/Ks ratios <1. These duplication events occurred around

**Table 2** Ka/Ks analysis of duplicate gene pairs in *DoEXP*s.

| Seq_1 | Seq_2 | Ka | Ks | ka/ks | Date (Mya) |
|---|---|---|---|---|---|
| *DoEXPA14* | *DoEXPA15* | 0.348976301 | 1.465376447 | 0.238147885 | 11.2721 |
| *DoEXLA1* | *DoEXLA2* | 0.180427892 | 0.512222507 | 0.352245148 | 3.9402 |
| *DoEXPA17* | *DoEXPA2* | 0.117321185 | 0.991671406 | 0.118306512 | 7.6282 |
| *DoEXPA19* | *DoEXPA2* | 0.147238695 | 0.857937152 | 0.171619441 | 6.5995 |
| *DoEXPA2* | *DoEXPA4* | 0.174745763 | 1.526510634 | 0.11447399 | 11.7424 |
| *DoEXPA17* | *DoEXPA4* | 0.199822805 | 1.474019019 | 0.135563247 | 11.3386 |
| *DoEXPA21* | *DoEXPA5* | 0.124372587 | 2.309011017 | 0.053864008 | 17.7616 |
| *DoEXPA5* | *DoEXPA7* | 0.095810531 | 1.277029241 | 0.075026106 | 9.8233 |
| *DoEXPA10* | *DoEXPA6* | 0.088856737 | 0.803862941 | 0.110537172 | 6.1836 |
| *DoEXPA18* | *DoEXPA6* | 0.077569824 | 0.88546908 | 0.087603086 | 6.8113 |
| *DoEXPA6* | *DoEXPA8* | 0.134544473 | 1.020308806 | 0.131866424 | 7.8485 |
| *DoEXPA21* | *DoEXPA7* | 0.116288398 | 1.559924484 | 0.074547453 | 11.9994 |
| *DoEXPA10* | *DoEXPA8* | 0.135464572 | 0.929768165 | 0.14569715 | 7.1521 |
| *DoEXPA18* | *DoEXPA8* | 0.120814847 | 0.991560743 | 0.121843112 | 7.6274 |
| *DoEXPA10* | *DoEXPA18* | 0.079365162 | 1.004022715 | 0.079047178 | 7.7233 |
| *DoEXPA12* | *DoEXPA20* | 0.176894509 | 1.146131211 | 0.154340539 | 8.8164 |
| *DoEXPA17* | *DoEXPA19* | 0.159389593 | 0.969968684 | 0.164324473 | 7.4613 |

3.9–17.8 million years ago (Mya). The results of the interspecific evolutionary relationships are shown in Fig. 7. A collinearity analysis comparing yam with Arabidopsis and rice was conducted. Twenty-six pairs of orthologous genes were identified between rice and yam. Although these 26 homologous gene pairs are collinear, the positions of the genes on the chromosomes differ slightly between yam and rice. Thirty-nine pairs of homologous genes were identified between yam and Arabidopsis. The Ka, Ks, and Ka/Ks values of the homologous gene pairs between yam and rice and between yam and Arabidopsis were subsequently calculated (Tables S3 and S4). The Ka/Ks values of the homologous gene pairs that could be calculated were all less than 1.0 (Fig. 8). The above results indicate that homologous genes of the expansin gene family were strongly subjected to purifying selection in yam with rice and Arabidopsis, as well as within the yam species.

## Analysis of *DoEXP*s expression

Using transcriptome data, the expression levels of *DoEXP*s at different time points were visualized in a heatmap. Most notably, the expression pattern of the *DoEXPA19* gene across the four different periods demonstrated marked divergence from those of other genes, and its expression peaked during the late expansion stage; thus, this gene was clustered into a distinct category (Fig. 9). The remaining genes were grouped into two categories: nine genes exhibited the highest expression in the expansion-formation period, with their expression levels generally downregulated thereafter; 18 genes displayed peak expression in the early expansion phase, and their expression patterns were subdivided into four subclasses. Notably, *DoEXLA2* exhibited an initial increase in expression followed by a decline, but the decrease only commenced during the late swelling stage. Minimal

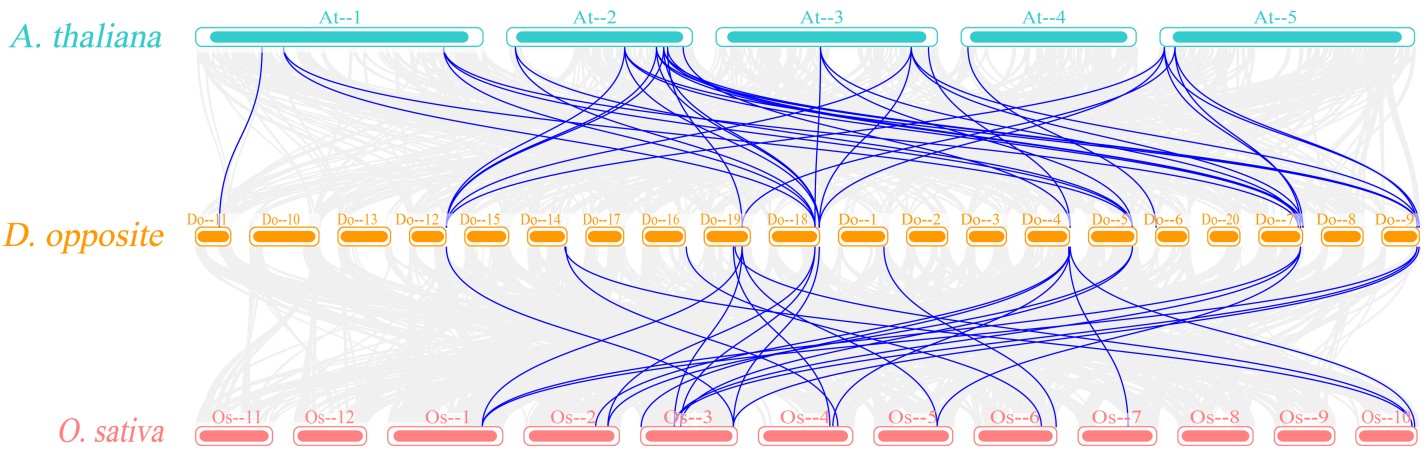

**Figure 7 Synteny analyses of *DoEXPs* to Arabidopsis and rice.** Gray background lines represent all collinear relationships between yam and the reference genomes of Arabidopsis and rice, while the blue lines specifically highlight the collinear gene pairs identified within the expansin gene family. Different color blocks represent the chromosomes of different species.

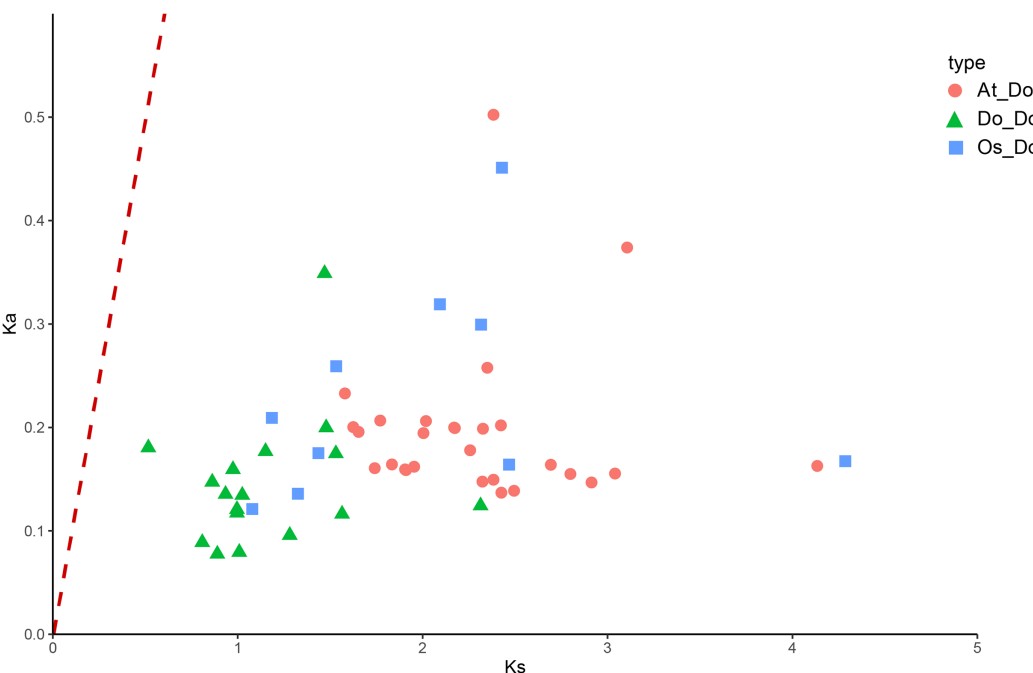

**Figure 8 The Ka/Ks values of all collinear gene pairs within and between species.** Distribution of Ka/Ks of all homologous gene pairs within yam (Do-Do), between yam and rice (Os-Do), and between yam and *Arabidopsis thaliana* (AT-Do). They are represented by green triangles, blue squares and pink circles respectively, and the red dotted line indicates the slope where Ka/Ks = 1.

variation in its expression was observed between the early and middle stages of swelling. The expression levels of ten genes including *DoEXPA1*, *DoEXPA3*, *DoEXPA4*, *DoEXPA5*, *DoEXPA8*, *DoEXPA10*, *DoEXPA12*, *DoEXPA16*, *DoEXPA18*, and *DoEXPB2* decreased during the middle stage of tuber enlargement and reached their lowest levels in the late

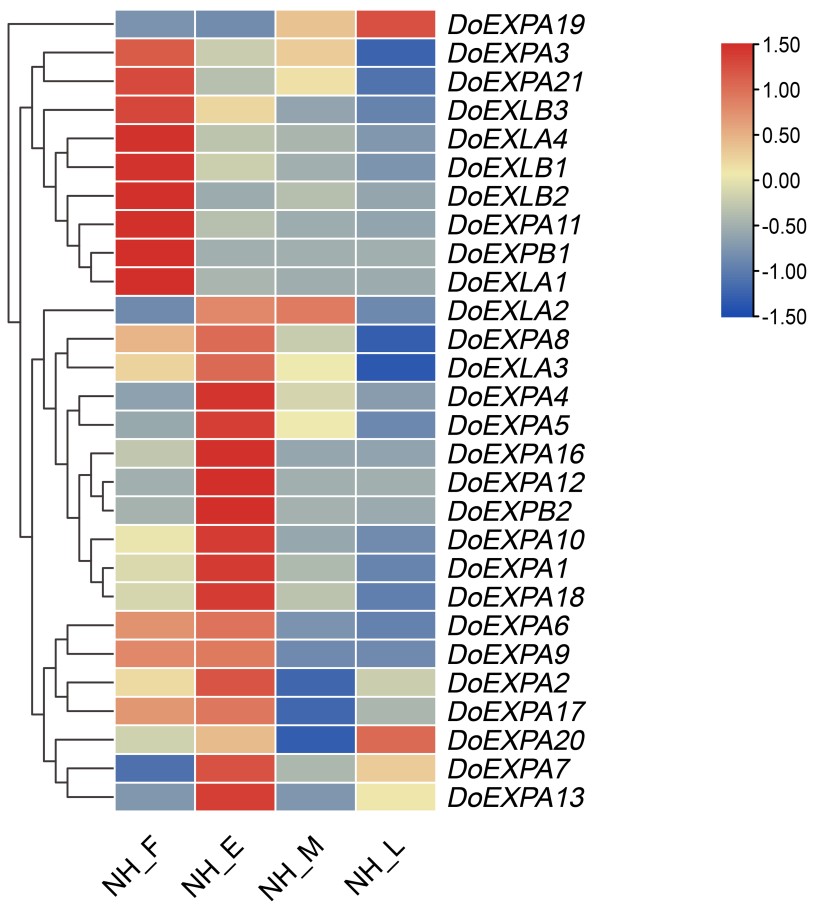

**Figure 9 Heat map analysis of the expression of *DoEXPs*.** NH_F/E/M/L represents the four tuber development periods of NH1 (abbreviated as NH). F, tuber formation stage; E, early tuber expansion; M, mid-tuber expansion; and L, late tuber expansion.

stage. Three genes, *DoEXPA7*, *DoEXPA13*, and *DoEXPA20*, exhibited expression patterns characterized by initial upregulation, subsequent downregulation, and a final upregulation, with pronounced differences across the four stages. The expression levels of four genes, *DoEXPA2*, *DoEXPA6*, *DoEXPA9*, and *DoEXPA17*, first tended to increase but then tended to decrease. Their expression levels remained consistently high during the swelling formation and early stages but were uniformly low during the middle and late stages of swelling. Moreover, 15 of the 18 genes (83.3%) exhibiting high expression levels during the early swelling stage belong to the EXPA subfamily. Genes in the *DoEXLB* subfamily showed relatively high expression during the tuber formation stage, but gradually decreased thereafter. These results suggest that the expression trends of *DoEXP*s are largely conserved among genes within the same subfamily.

## GO functional enrichment analysis of *DoEXP*s

The functions of *DoEXP*s were predicted by GO enrichment analysis, which categorized them into three modules, including cellular component (CC), biological process (BP), and molecular function (MF). These genes were associated with 16 GO terms, including one

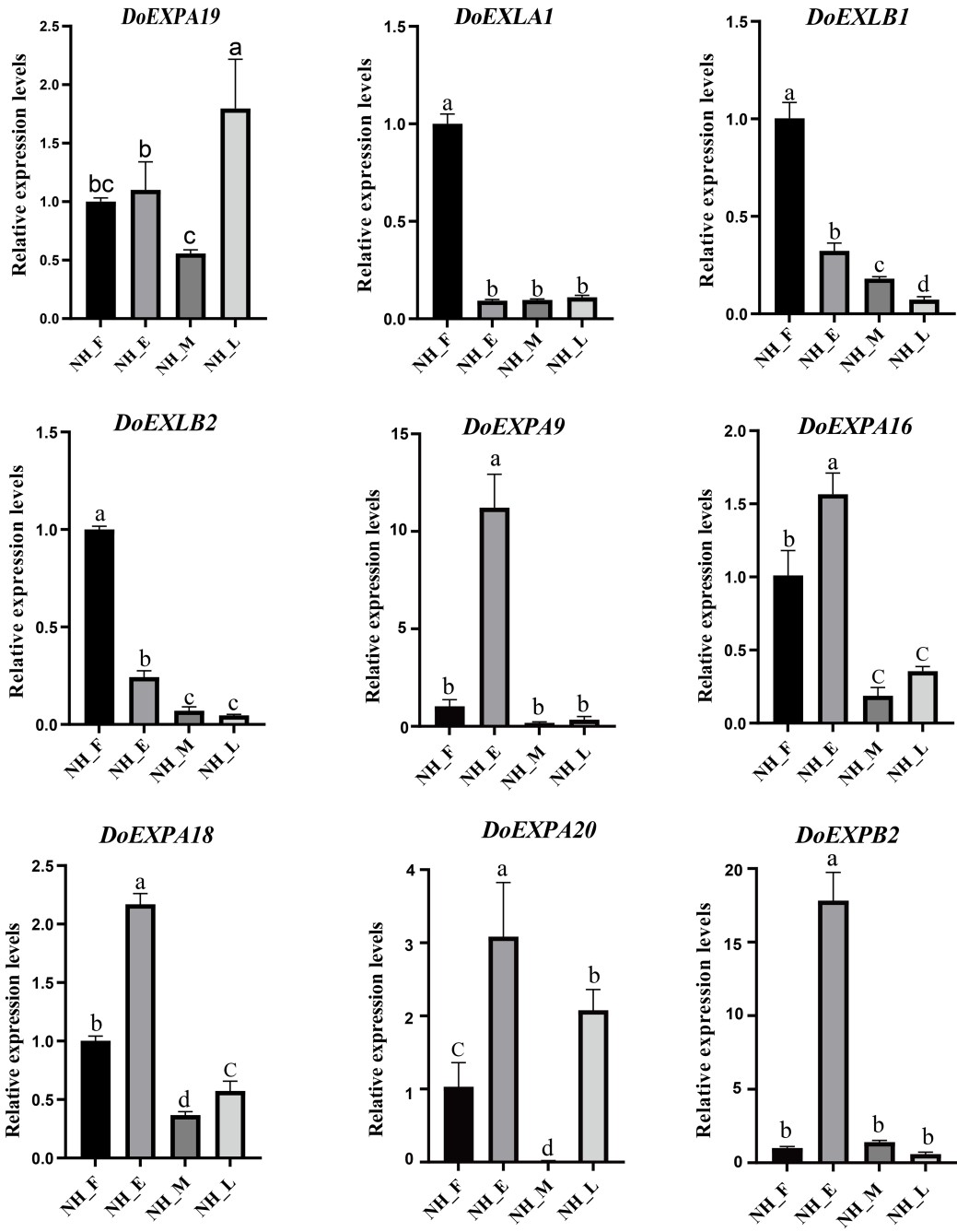

**Figure 10 qPCR expression analysis of nine selected *DoEXP*s in different periods.** Tuber tissues of yam were sampled at four stages (NH_F/E/M/L) for expression analysis. Three biological replicates were performed and the data are expressed as mean ± SD. Data analysis was performed separately for each of the nine panels. Different lowercase letters indicate the results of multiple comparisons based on a *post-hoc* test (Duncan's test). Bars assigned the same letter are not statistically significantly different, whereas those labeled with different letters differ significantly at the significance level of *P* < 0.05.

term under CC, 15 terms under BP, and no significantly enriched terms under MF
(Fig. S1). The enriched entries in the CC category represented plant cell wall, consistent
with their predicted cell wall localization in our earlier analysis. The biological processes
primarily involved growth and development of the plant root system and morphogenesis.

### RNA-Seq data validation

To validate the reliability of RNA-Seq data, several DoEXP genes were randomly selected
from each group. Their expression was then verified *via* qPCR (Fig. 10). All primers used
in qPCR analysis are listed in Table S5. Expression patterns of the nine transcripts during
tuber development were consistent with RNA-Seq results, despite discrepancies in fold
changes, thereby confirming the reliability of our RNA-Seq data and subsequent analysis.

## DISCUSSION

### Overview of the cell wall composition and expansins

The plant cell wall is a unique structure essential for plant growth, cell differentiation, and
intercellular communication. Growing plant cells are enclosed by a primary cell wall, a
heterogeneous matrix composed of cellulose microfibrils non-covalently linked to
hemicellulose, pectin, and structural proteins. This structure dictates cell morphology and
dynamically adjusts during growth to accommodate cell expansion, a process pivotal for
plant development (*Cosgrove, 1997*, *2005*; *Zhong & Ye, 2015*; *Cosgrove, 2024a*).
Biochemical characterization of the "acid growth" mechanism in cell walls led to the
discovery of expansins, which disrupt noncovalent bonding between microfibrils and the
matrix (*Shieh & Cosgrove, 1998*). They belong to a large gene superfamily encoding small
proteins with 225–300 amino acid residues (*Fukuda, 2014*). The main expansin families in
plants include α-expansin (EXPA), which targets cellulose−cellulose linkages; β-expansin
(EXPB), which acts on xylan (a major hemicellulose component); and expansin-like A/B
(EXLA/B) (*Kende et al., 2004*). The auxin signal activates the plasma membrane $H^+$-
ATPase, reducing the cell wall pH and enhancing cell wall extensibility (*Du, Spalding &
Gray, 2020*). EXPA mediates acid-induced growth, enabling cell wall expansion driven by
hormones and growth factors (*Cosgrove, 2024b*). EXPB, classified as a Group 1 grass pollen
allergen (Bet v 1 family), is highly expressed in grass pollen and may facilitate cell wall
loosening during pollen tube growth toward the ovary (*Sampedro & Cosgrove, 2005*;
*Li, Chang & Guan, 2022*). EXLA/EXLB has also been implicated in seed and root
development, as well as stress resistance processes in various plants (*Boron et al., 2015*;
*Kong et al., 2019*).

### Identification and functional analysis of *DoEXP*s

In this study, whole-genome analysis identified 30 expansin genes with two conserved
domains in yam. These genes (*DoEXP*s) were classified into four subfamilies (EXPA,
EXPB, EXLA, and EXLB), containing 21, 2, 4, and 3 members, respectively (Table 1).
Chromosomal distribution analysis showed that *EXPs* were distributed across 14
chromosomes, and genes were renamed as *DoEXPA1-21*, *DoEXPB1-2*, *DoEXLA1-4*, and
*DoEXLB1-3*. Analysis of conserved motifs and gene structures revealed high structural

conservation within subfamilies, suggesting potential functional similarities. All *DoEXPs* contained conserved motifs 1 and 6, critical for EXP gene family evolution. Structural motif analysis further confirmed consistent intron/exon numbers within each subfamily, and that the phylogenetic relationship between the EXLA and EXLB subfamilies was closer. The positions of conserved domains align with prior studies (*Kende et al., 2004*). Cis-acting elements analysis in the 2,000 bp promoter region of *DoEXPs* revealed 20 genes harboring light-responsive elements, suggesting involvement in photomorphogenesis regulation. These findings were consistent with the results reported in wheat, where EXP genes respond to light signals (*Yang et al., 2023*). In addition, the *EXPs* are involved in multiple hormone response pathways, with 37 ABA-responsive elements detected— among the highest across stress-related hormones. ABA acts as a key endogenous messenger under abiotic stresses (*e.g.*, drought, low temperature, and salinity), transmitting signals to downstream pathways (*Raghavendra et al., 2010*; *Lee & Luan, 2012*; *Chen et al., 2019*; *Zhang et al., 2025b*). This supported the hypothesis that EXP genes mediate plant responses to ABA and other hormones. Overexpression studies in cotton and wheat confirmed *EXPs* enhance stress tolerance (*Chen et al., 2017*; *Zhang et al., 2021*). Moreover, 31 gibberellin (GA) response elements were identified. Our previous study demonstrated that the expression levels of five *DoEXPs* were significantly up-regulated under GA3 treatment (*Zhou et al., 2021*). Additionally, 16 salicylic acid (SA) response elements were detected, suggesting that EXPs may be involved in SA-mediated defense processes (*Muthusamy et al., 2020*). A total of 12 auxin response elements were identified in *DoEXPs*, with eight located in *DoEXPAs*. As EXPA mediates acid growth at pH < 5 (*Durachko & Cosgrove, 2009*), this subfamily's prevalence in growth-related processes is biologically plausible. These results highlighted the DoEXPA subfamily's role in root growth and morphogenesis. Detailed analysis of *cis*-element composition in individual genes revealed that *DoEXPA9*, *DoEXPA16*, *DoEXPA18*, and *DoEXPA20* possessed ≥10 active elements. *DoEXPA16* and *DoEXPA20* exhibited the most diverse elements, including light-, hormone-, and abiotic stress-responsive types. *DoEXPA18* contained 13 elements, dominated by light-, ABA-, and MeJA-responsive signals. *DoEXPA9* showed drought- and cold-responsive elements, suggesting that this gene had a relatively strong ability to resist external abiotic stresses. These findings suggested that *DoEXPs* can function under abiotic stress, which has been confirmed in plants such as poplar, willow, and tobacco (*Marowa et al., 2020*; *Yin et al., 2023*; *Zhang et al., 2025a*).

## Evolution and trends in the expression of *DoEXPs*

The evolution of gene families is primarily driven by tandem and segmental duplications (*Kent et al., 2003*). To investigate the evolutionary relationships of the DoEXP gene family, we analysed its duplication patterns. A total of two tandem duplication pairs and 16 segmental duplication pairs were identified, indicating that segmental duplication was the predominant mechanism shaping the evolution of *DoEXPs*. This result is consistent with previous studies on expansin gene family evolution in diverse species (*Zhu et al., 2014*; *Han et al., 2019*; *Li et al., 2021*). Ka/Ks ratio analysis of duplicated DoEXP gene pairs and

collinear EXP homologous genes in rice and Arabidopsis revealed that the EXP gene family underwent purifying selection throughout evolution.

Transcriptome analysis identified three major expression patterns of *DoEXP*s during the four developmental stages. *DoEXPA19* exhibited the highest expression during the late swelling stage. The second group of genes peaked during the tuber formation stage, and their expression levels gradually declined to the lowest in the late swelling stage. The genes in other groups presented the highest expression in the early swelling stage and then decreased gradually. Given the highest tuber swelling rate occured from early to mid-swelling stage (*Gong et al., 2017*), the peak expression of *DoEXP*s during these phases suggested their critical role in regulating tuber expansion. Among them, 18 genes with highest expression levels in the early swelling stage were suggested to play important roles in tuber swelling regulation.

Functional analysis further revealed that *DoEXPA9*, *DoEXPA16*, *DoEXPA18*, and *DoEXPA20* contained abundant and diverse *cis*-acting elements, coinciding with their highest expression levels in the early swelling stage. This characteristic was also verified by qPCR, which suggested that these four genes could serve as priority candidates for investigating *DoEXP*-mediated tuber swelling regulation, as well as their potential roles in hormone signalling and abiotic stress tolerance in plants.

## CONCLUSIONS

Overall, this study identified and analysed the genes of the expansin family in yam at the whole-genome level for the first time. A total of 30 expansin genes were identified, and their physicochemical properties and evolutionary relationships were analysed. The results revealed that expansin genes play important roles in abiotic stress responses as well as root growth and development. These findings advance the understanding of expansin functions in yam, providing a foundation for future research on tuber development and stress adaptation.

### Funding

This work was funded by the Natural Science Foundation of China (Grant No. 32060419) and the earmarked fund for CARS(CARS-21). The funders had no role in study design, data collection and analysis, decision to publish, or preparation of the manuscript.

### Grant Disclosures

The following grant information was disclosed by the authors:
Natural Science Foundation of China: 32060419.
The earmarked fund for CARS(CARS-21).

### Competing Interests

The authors declare that they have no competing interests.

## Author Contributions

- Xing Qiao performed the experiments, analyzed the data, prepared figures and/or tables, authored or reviewed drafts of the article, and approved the final draft.
- Shuai Yuan performed the experiments, prepared figures and/or tables, established a self-built Gene Ontology (GO) library for the research species and analyzed the results, and approved the final draft.
- Jing Wei performed the experiments, prepared figures and/or tables, and approved the final draft.
- Changzhou Li analyzed the data, prepared figures and/or tables, provided the analysis software and its usage instructions, and approved the final draft.
- Lan Lan analyzed the data, prepared figures and/or tables, and approved the final draft.
- Xuerui He performed the experiments, prepared figures and/or tables, provided the experimental materials required for the research, and approved the final draft.
- Aiqin Wang analyzed the data, authored or reviewed drafts of the article, and approved the final draft.
- Jie Zhan analyzed the data, authored or reviewed drafts of the article, and approved the final draft.
- Longfei He conceived and designed the experiments, authored or reviewed drafts of the article, and approved the final draft.
- Dong Xiao conceived and designed the experiments, authored or reviewed drafts of the article, coordinated and managed the progress of the research project, and approved the final draft.

## Data Availability

The data is available at Figshare: Qiao, Xing (2025). Identification and Analysis of the Expansin gene family in Yam. figshare. Dataset. https://doi.org/10.6084/m9.figshare.28473086.v2.

Raw data is available in the Supplemental File.

## Supplemental Information

Supplemental information for this article can be found online at http://dx.doi.org/10.7717/peerj.20093#supplemental-information.

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
