# Peer review of "Identification and analysis of the expansin gene family in yam"

_PeerJ, doi:10.7717/peerj.20093_

## Round 0.1 · original submission · Major Revisions

·

Basic reporting

This study comprehensively identified expansin genes in yam plants and provided some critical structural, biochemical, and functional data. Generally, all the parts of this manuscript are well written and presented. As the reader is capable of getting enough basic data about the role of expansins in yam plants and their role in tuber development. This is very crucial for future studies aimed at improving the yield (tuber growth) of yam plants.

Experimental design

Sufficiently presented, and according to the general rules of equivalent experiments. All the experiments are comprehensive.

I suggest that the author also could do a quantitative PCR on yam plants under various abiotic stresses and hormonal treatments.

Validity of the findings

In general, the results are well presented and fully explained. The statistical analysis is standard and sound.

Additional comments

Please check the attached revised MS for some minor corrections and suggestions.

Reviewer 2 ·

Basic reporting

This study characterizes the DoEXP gene family in Dioscorea opposita, a crop of significant agricultural importance. But this MS is full of flaws and errors. I would suggest that the authors rewrite the manuscript and omit vague terms. Avoid double text and sentences. Scientifically, the data is not well presented either and needs restructuring. Only after the data and writing are improved can the conclusions be assessed to see if they are justified. The paper is, in my view, not ready for peer review, and I recommend serious rewriting and restructuring before reconsideration.

Experimental design

Scientifically, the data is not well presented either and needs restructuring.

Validity of the findings

Only after the data and writing are improved can the conclusions be assessed to see if they are justified.

Additional comments

The paper is, in my view, not ready for peer review, and I recommend serious rewriting and restructuring before reconsideration.

·

Basic reporting

The abstract needs to be rewritten following the points below in a sequential order;
1. Brief introduction (written very well already)
2. Problem statement (written very well)
3. Justification of the study (written very well)
4. Methodology (written very well)
5. Results (There were mixtures of methodology in the results section of the abstract)
6. Conclusion (written very well).

Experimental design

Well designed

Validity of the findings

-

Additional comments

The research is well thought out and written soundly. However, the abstract should convey the message clearly if adjusted as recommended above. All over the work, the oldest references should be written first, followed by the recent citations. Correct this all over the work.

---

## Round 0.2 · Major Revisions

Still some major revisions are required to accept your article. Kindly revise thoroughly accordingly.

**Language Note:** The review process has identified that the English language must be improved. PeerJ can provide language editing services - please contact us at [email protected] for pricing (be sure to provide your manuscript number and title). Alternatively, you should make your own arrangements to improve the language quality and provide details in your response letter. – PeerJ Staff

·

Basic reporting

The response of the authors to the comments of the first revision is accepted.

Experimental design

According to the rules of scientific research.

Validity of the findings

They are well presented and show valuable data about the biology and function of expansins in yam plants.

Reviewer 2 ·

Basic reporting

This study provides valuable genomic resources for yam expansins and lays the groundwork for future functional studies, but there are some issues I must highlight before considering for publication. I recommend a major revision to address these issues before reconsideration. It is strongly recommended to authors that they should perform a careful line-by-line review for consistency and clarity. The manuscript is full of typo errors and grammatical mistakes, e.g., some of them.
Please consider modifying it, e.g.,
Line 58: In this research
Line 141 was exploded to identify
Line 258: The EXPA subfamily is the largest group,
Line 380: acid-growth" → "acid growth"
Line 368: quantitative real-time PCR (qPCR) on randomly selected DoEXPs (Fig.10).
"Fig.10" → "Fig. 10" (space required), please use a consistent writing style throughout the text.
Line 217: for each stage. The samples were immediately flash-frozen in liquid nitrogen and then stored (remove extra space).
Line 353-354: Moreover, among the 18 genes with high expression levels at the early swelling stage, 15 (83.3%) belong to the EXPA subfamily. Please check the grammar again.
Line 286: A total of 13 major cis-acting elements were detected, including light-, hormone-
I see there are 11 Figure files (s) info, but I can't find fig. 11 in the text.
What is the rationale for selecting the 9 genes for qPCR validation?
Line 347: levels in the late stage. Three genes ( DoEXPA7, DoEXPA13, and DoEXPA20 ) exhibited what is special about writing these genes in brackets, but other genes do not.
Line 368: quantitative real-time PCR (qPCR), why authors using the full text again and again if mentioned?
Please address and rewrite the methodological ambiguities in the sampling and qPCR sections.

Experimental design

Please address and rewrite the methodological ambiguities in the sampling and qPCR sections.

Validity of the findings

The identification of 30 genes, their classification, characterization of conserved domains/motifs/exon-intron structures, promoter analysis, and insights into duplication events form a solid foundation for understanding expansin function in yam.

Additional comments

This manuscript presents a timely and valuable bioinformatics analysis identifying and characterizing the expansin gene family (DoEXP) in yam.

---

## Round 0.3 · accepted · Accept

Thanks for following the review comments and your manuscript now is ready for publication.

Reviewer 2 ·

Basic reporting

Accept

Experimental design

Accept

Validity of the findings

Accept

Additional comments

Accept